# Functional Gastrointestinal Disorders in Outpatients Aged up to 12 Months: A French Non-Interventional Study

**DOI:** 10.3390/ijerph17114031

**Published:** 2020-06-05

**Authors:** Florence Campeotto, Marie-Odile Barbaza, Veronique Hospital

**Affiliations:** 1Pediatric Gastroenterology Department, Necker-Enfants-malades Hospital (AP-HP), 75015 Paris, France; 2Microbiological Laboratory, Faculty of Pharmacy, Inserm U1139, University of Paris, 75006 Paris, France; 3Auxesia, 69150 Décines, France; marie-odile.barbaza@auxesia.net; 4Medical Departement, Nutrition Hygiène Santé, Menarini Group, 94150 Rungis, France; departement.medical@menarini.fr

**Keywords:** colic, constipation, functional diarrhea, infant, regurgitation, prevalence

## Abstract

Background: The aim of this study was to estimate the frequency of functional gastrointestinal disorders (FGIDs) in infants aged up to 12 months according to the new ROME IV criteria defining these disorders, and to describe the management of FGIDs in France. Methods: This French non-interventional, cross-sectional, and multicenter study was conducted among private-outpatient physicians who each included four consecutive patients aged up to 12 months. The frequency of FGIDs was described using the ROME IV criteria versus clinicians’ diagnosis. The characteristics of infants with and without FGID were compared, and the management of the FGIDs was described. Results: In the 1722 infants analyzed, the following frequencies were observed according to the ROME IV criteria versus the physicians’ diagnosis: regurgitation 41% versus 45%; colic 18% versus 30%; constipation 9% versus 19%; diarrhea 3% versus 8%. Of note, FGID infants were less frequently exclusively breastfeeding at the maternity hospital (*p* < 0.001), were introduced to cow’s milk earlier after leaving the maternity hospital (*p* < 0.001), and more frequently had symptoms suggestive of cow’s milk protein allergy (*p* < 0.001). Physicians frequently recommended an adapted infant formula (in 77% to 82% of cases depending on the FGID diagnosed) and prescribed a specific treatment in 51% to 66% of infants (probiotics: 35% to 64%). Conclusions: This real-world study confirms the high frequency of FGIDs in infants in France, and provides new information regarding the characteristics of FGID infants.

## 1. Introduction

Functional gastrointestinal disorders (FGID), mainly regurgitation, colic, constipation, and diarrhea, are frequent in infants [1] and lead to numerous visits to the pediatrician and the general practitioner (GP). However, little is known about the real frequency of these disorders in infants aged up to 12 months. A literature review published in 2005 [2] reported very heterogeneous prevalence data in this age group. For most FGIDs, many of the studies referenced were based on very old data; furthermore, study designs, the populations studied, and definitions of the symptoms made it difficult to draw firm conclusions. Consequently, based on a questionnaire sent to practicing clinicians worldwide, an international expert group agreed in 2015 on the following probable prevalence estimates for infants aged up 12 months [2]: 30% for regurgitation, 20% for colic (including crying and fussing), 15% for constipation, and less than 10% for diarrhea. The overall prevalence estimate for FGIDs was up to 30% in this age group. Practical international and local recommendations are available for managing FGID in infants [3,4,5,6].

Considering the still uncertain prevalence estimates for FGID in infants, as well as the lack of data for France, and the release of the ROME IV criteria in May 2016 [5,6], this French non-interventional study was mainly designed to estimate the frequency of functional regurgitation, colic, diarrhea, and constipation in infants aged up to 12 months, in private-outpatient practice medicine. The study also aimed to describe the management of these symptoms in a real-life setting.

## 2. Materials and Methods

This French non-interventional, cross-sectional, and multicenter study was conducted between October 2016 and September 2017 among private-outpatient pediatricians and GPs involved in infant health management. The study was compliant with French requirements regarding the protection of privacy and personal data. Infants’ legal representatives signed a written informed consent prior to study participation. The study protocol was approved by the French National Medical Council.

### 2.1. Recruitment of Physicians and Patients

To minimize selection bias, physicians with a private practice were randomly selected from an updated French national database, using the quotas method to ensure geographical representativeness in the recruitment. Each physician, during the study period, was asked to consecutively include four outpatient infants aged up to 12 months, and for whom the parental consent was obtained (regardless of the reason for the medical visit). In order to ensure sufficient data in each of the predefined age subgroups, physicians had to include two infants aged up to 6 months and two infants aged more than 6 months. Infants participating in a clinical trial were excluded from the study.

### 2.2. Data Collection

The age, gender, specialty, type of medical practice, and geographic location of participating physicians were collected. Their geographical distribution was compared with that of the 61,719 French community-based pediatricians and GPs [7]. At inclusion, physicians collected infant characteristics, clinical symptoms suggestive of cow’s milk protein allergy and family history of atopy, feeding at the maternity hospital, and the presence of functional regurgitation, colic, diarrhea, and/or constipation according to the physician’s opinion, with the detailed symptoms of each FGID including the Vandenplas codification in the case of regurgitation [8]. The management of each FGID was also collected (type of infant formula recommended and/or treatments prescribed when applicable).

### 2.3. Study Size and Statistical Methods

At the time of the study, no prevalence data using the recent ROME IV criteria [6] were available for FGID in infants. In 2015, a study conducted in the US on infants under the age of 12 months, and based on the ROME III criteria, showed that the least frequent FGID was diarrhea (3%). On the basis of this, an overall sample size of around 2000 infants aged up 12 months made it possible to describe this proportion of FGID (relative precision of ±25%, 95% confidence interval (CI)).

The frequency of each FGID in infants aged up to 12 months and followed as outpatients by community-based pediatricians and GPs was described using the ROME IV criteria, with the associated 95% CI (Table 1).

These proportions were also described according to the physicians’ own diagnosis. For each FGID, the degree of agreement or concordance between the diagnosis by clinicians and the ROME IV criteria was measured using the Cohen’s kappa coefficient (value between 0 and 1) [9]. The infants’ characteristics and their FGID symptoms as defined by the ROME IV criteria were also described, as well as the management of digestive symptoms (infant formula and treatments). Two-sided tests with type I error α = 0.05 were applied when relevant and were considered significant at *p* < 0.05. Comparisons between infants with and without FGID were carried out using the usual tests (Chi-square or Fishers exact test for the qualitative variables and the Student *t*-test or Wilcoxon test for the quantitative variables). Missing data were not replaced. Statistical analyses were performed using SAS^®^ software, version 9.2 (SAS Institute Inc., Cary, NC, USA).

## 3. Results

### 3.1. Physician and Patient Populations

Almost all of the 451 participating physicians were pediatricians (91%; GPs: 9%). They were distributed throughout France, with more clinicians in the Ile-de-France area compared with our source population (25% versus 16% throughout France). The participating physicians were mainly women (67%) and they practiced in urban areas in 83% of the cases.

Of the 1781 infants included, 59 (3%) were excluded from analysis (30 for missing or incorrect visit date and/or 29 for no parental consent) leading to an analysis population of 1722 patients. Of these 1722 infants, 997 (58%) were aged up to 6 months.

The presence or otherwise of FGID(s) was not established by physicians for 30 infants (2%). Overall, 1154 (67%) suffered from at least one FGID according to the clinician’s diagnosis. In the 1321 infants who could be assessed using the ROME IV criteria (77% of patients from the analysis population, Table 1), there were 741 with at least one FGID (56%).

### 3.2. Characteristics and Feeding of Infants

Infants and feeding characteristics are detailed in Table 2. Infants with at least one FGID were younger at study visit compared with those without a FGID (*p* < 0.001); they were more often boys (*p* = 0.013), and had a lower weight at birth (*p* = 0.019) with more frequent failure to thrive (*p* = 0.012). In addition, they presented more often with symptoms suggestive of cow’s milk protein allergy (*p* < 0.001). At the maternity hospital, exclusive breastfeeding was less frequent in infants with FGID(s) (*p* < 0.001). Of the 583 infants (35%) who were introduced to cow’s milk after leaving the maternity hospital, (with FGID(s): 33%; without FGID: 38%), it was done earlier in infants with FGID(s) (*p* < 0.001).

### 3.3. Frequency of FGIDs and Symptoms Related to FGIDs

According to the ROME IV criteria (Table 1), regurgitation was the most frequent FGID observed (41%), followed by colic, constipation and diarrhea (Table 3). These proportions were lower than those estimated by physicians. The concordance between the diagnosis by clinicians and the ROME IV criteria was moderate for regurgitation (kappa coefficient: 0.55, CI 95% (0.51–0.59)) and constipation (0.54, CI 95% (0.48–0.59)) and poor for colic (0.39, CI 95% (0.35–0.44)) and diarrhea (0.24, CI 95% (0.15–0.33)). The infants’ symptoms reported by physicians for each FGID are detailed in online Appendix A.

### 3.4. Management of FGIDs

Regardless of the type of FGID observed, the physicians recommended a baby-milk formula they considered best suited to the digestive disorder for most infants (in between 77% and 82% of infants, depending on the FGID diagnosed) (Table 4). However, a thickening, anti-reflux or comfort formula was frequently recommended (regurgitation: 70% of infants concerned; colic: 45%; constipation: 35%; diarrhea: 26%). In addition, physicians prescribed a specific treatment for between 51% and 66% of infants, depending on the FGID diagnosed (alginate/gastric protectants, probiotics or laxatives in particular). It should be noted that probiotics were often prescribed regardless of the FGID (regurgitation: 35%; colic: 54%; constipation: 36%; diarrhea: 64%).

## 4. Discussion

This large French non-interventional study conducted in 2017 on the basis of the new ROME IV criteria [5,6] confirmed previous findings showing the high frequency of FGIDs in infants up to the age of 12 months, in current clinical practice [1,2].

By comparison with data based on the ROME III criteria [1] from a recent multinational literature review and an expert consensus [2], the frequency estimation was higher for regurgitation (41% of infants using ROME IV criteria, versus 30%) and fairly close for other FGIDs (colic: 18% versus 20%; constipation: 9% versus 15%; diarrhea: 3% versus <10%).

In this French study conducted in a real-life setting, most of the infants with FGIDs were specifically managed by clinicians with at least one dedicated baby-milk formula in more than three quarters of the cases, and at least one drug prescription in at least half of the cases regardless of the type of FGID.

In addition, the study showed that in infants suffering from FGID(s), exclusive breastfeeding was less common at the maternity hospital, and cow’s milk was introduced earlier after leaving the maternity. Differences between infants with and without FGIDs were significant. However, the causal relationship between both parameters remains unclear: persistent FGIDs when being breastfed could lead to a change to a special formula, or the higher prevalence of formula feeding after the first week of life and the earlier initiation of formula in the FGID group could reflect the physician’s effort to improve the symptoms of an emerging FGID.

In our study, clinicians frequently prescribed probiotics for infants with a FGID (from 35% of the infants treated for regurgitation up to 64% of the infants treated for diarrhea), suggesting that they hoped that probiotics could improve the digestive disorders. The settlement of the gut microbiota begins quickly in newborns. The pattern of bacterial colonization is in particular affected by the mode of delivery, environmental conditions, gestational age, antibiotic administration, and the type of feeding, and may result in dysbiosis [10,11]. In breast-fed infants, the gut microbiota is less diversified than in formula-fed infants, with a predominance of *Bifidobacterium* (a genus known for its potential health benefits) [11]. Previous studies have shown that supplementation with probiotic strains increases the intestinal colonization by *Bifidobacterium* in full-term and premature infants [12,13], and was effective in managing infant colic [14]. In addition, the level of *Bifidobacterium* colonization is lower in allergic infants than in non-allergic infants [15]. In our study, symptoms suggestive of cow’s milk protein allergy were more frequent in infants with FGID(s). Altogether, the potential benefits provided by supplementation with bifidobacteria could explain the frequency of prescriptions of probiotics in FGID infants. However, in this cross-sectional study, it was not possible to assess the impact of probiotics on FGIDs.

With regard to the limitations of this study, a priori procedures were set up to minimize selection bias of both physicians and patients. However, most of the participating physicians were pediatricians leading to a potential selection bias as the infants they included may have been suffering from more severe disorders than those managed by GPs. In addition, the digestive symptoms studied were collected through parent reports to clinicians and were not based on detailed dairies. On the other hand, data collection based on detailed diaries could have led to another potential selection bias in the context of a real-life study: parents agreeing to complete detailed diaries would probably be the most anxious ones, with the most affected infants (leading to an overestimation of the prevalence of FGIDs).

## 5. Conclusions

This large French real-world study confirms and updates previous estimations for FGID frequency in infants, and provides new information regarding the characteristics of the infants who developed FGIDs. Further studies are needed to better assess the role of breastfeeding in the development and the management of FGIDs.

## Figures and Tables

**Table 1 ijerph-17-04031-t001:** Diagnosis criteria for functional gastrointestinal disorders in infants—ROME IV criteria.

Functional Gastrointestinal Disorders	ROME IV Criteria
Regurgitation	Must include both of the following in otherwise healthy infants 3 weeks to 12 months of age: Regurgitation 2 or more times per day for 3 or more weeks No retching, hematemesis, aspiration, apnoea, failure to thrive, feeding or swallowing difficulties, or abnormal posturing
Colic	For clinical purposes, must include all of the following: An infant who is <5 months of age when the symptoms start and stop Recurrent and prolonged periods of infant crying, fussing, or irritability reported by caregivers that occur without obvious cause and cannot be prevented or resolved by caregivers No evidence of infant failure to thrive, fever, or illness
Diarrhoea	Must include all of the following: Daily painless, recurrent passage of 4 or more large, unformed stools Symptoms last more than 4 weeks Onset between 6 and 60 months of age No failure to thrive if caloric intake is adequate
Constipation	Must include 1 month of at least 2 of the following in infants up to 4 years of age: 2 or fewer defecations per week History of excessive stool retention History of painful or hard bowel movements History of large-diameter stools Presence of a large faecal mass in the rectum

**Table 2 ijerph-17-04031-t002:** Characteristics and feedings of infants with and without functional gastrointestinal disorders (*N* = 1722).

	With FGID *N* = 1154	Without FGID *N* = 538	*p* Value
Age at study visit (months)—mean ± SD	4.92 ± 3.11	6.69 ± 3.19	<0.001
Boys—*n* (%)	636 (55.4)	260 (48.9)	0.013
Prematurity—*n* (%)	44 (4.3)	22 (4.4)	NS
Born by natural way—*n* (%)	936 (83.1)	449 (84.9)	NS
Birth body weight (kg)—mean ± SD	3.25 ± 0.48	3.31 ± 0.45	0.019
Body weight at study visit (kg)—mean ± SD	6.65 ± 1.90	7.59 ± 1.84	<0.001
General good health at study visit—*n* (%)	1012 (89.2)	470 (88.2)	NS
Failure to thrive—*n* (%)	49 (4.3)	10 (1.9)	0.012
Symptoms suggestive of CMPA—*n* (%)	99 (8.7)	12 (2.2)	<0.001
Family history of atopy—*n* (%)	267 (23.4)	122 (22.9)	NS
Feeding at the maternity hospital—*n* (%)			<0.001
Exclusive breastfeeding	530 (46.0)	297 (55.6)	
Breastfeeding + infant formula	164 (14.2)	51 (9.6)	
Infant formula only	458 (39.8)	186 (34.8)	
Feeding at study visit—*n* (%)			NS
Exclusive breastfeeding	151 (13.5)	60 (11.5)	
Breastfeeding + infant formula	155 (13.8)	67 (12.9)	
Infant formula only	814 (72.7)	394 (75.6)	
Introduction of cow’s milk—*n* (%)			0.071
Not yet (at the time of the study visit)	515 (45.7)	215 (41.0)	
During the first week of life (≈ at the maternity)	243 (21.6)	108 (20.6)	
After the first week of life	368 (32.7)	201 (38.4)	
Age when cow’s milk introduced after the first week of life (months)—mean ± SD	2.71 ± 1.90	3.39 ± 2.13	<0.001
Dietary diversification—*n* (%)	549 (48.5%)	377 (71.3%)	<0.001
Age of dietary diversification (months)—mean ± SD	4.61 ± 0.69	4.64 ± 0.69	NS

CMPA (cow’s milk protein allergy); FGID (functional gastrointestinal disorder); NS (non-significant); SD (standard deviation); Number of missing data (with FGID, without FGID): Age (*n* = 3, 2); Gender (*n* = 5, 6); Prematurity (*n* = 129, 39); Birth delivery (*n* = 27, 9); Birth weight (*n* = 12, 6); Weight at study visit (*n* = 37, 25); General health (*n* = 20, 5); Failure to thrive (*n* = 16, 2); Symptoms suggestive of CMPA (*n* = 16, 2); Family history of atopy (*n* = 15, 6); Feeding at maternity (*n* = 2, 4); Feeding at study visit (*n* = 34, 17); Introduction of cow’s milk (*n* = 28, 14); Age of cow’s milk introduction if after 1st week (*n* = 0, 0); Dietary diversification (*n* = 21, 9); Age of dietary diversification (*n* = 0, 0).

**Table 3 ijerph-17-04031-t003:** Frequency of functional gastrointestinal disorders in infants.

	According to the ROME IV Criteria * *n* (%) [95% CI]	According to Physicians Diagnosis ** *n* (%)
Regurgitation	516/1,256 (41.1) [38.4–43.8]	757/1688 (44.8)
Colic	171/946 (18.1) [15.6–20.5]	509/1706 (29.8)
Constipation	141/1,570 (9.0) [7.6–10.4]	323/1682 (19.2)
Diarrhea	21/753 (2.8) [1.6–4.0]	133/1689 (7.9)

CI (Confidence Interval); * Analysis was restricted to only the infants with no missing data and taking into account the specified age ranges defined by the ROME IV criteria: regurgitation: ≥3 months (*n* = 1697); colic: ≤6 months (*n* = 997); diarrhea: ≥6 months (*n* = 838); constipation: no age criterion (*n* = 1722); ** Missing data (results according to physicians’ diagnosis): regurgitation (*n* = 34); colic (*n* = 16); diarrhea (*n* = 33); constipation (*n* = 40).

**Table 4 ijerph-17-04031-t004:** Management of infants with functional gastrointestinal disorders.

	Regurgitation *N* = 757	Colic *N* = 509	Constipation *N* = 323	Diarrhoea *N* = 133
Infant formula recommendation—*n* (%)	615 (82.1)	384 (77.3)	249 (78.5)	105 (82.0)
Thickening, anti-reflux, comfort	433 (70.4)	171 (44.5)	88 (35.3)	27 (25.7)
Standard	62 (10.1)	39 (10.2)	50 (20.1)	22 (21.0)
Protein hydrolisate	59 (9.6)	41 (10.7)	14 (5.6)	23 (21.9)
Anti-colic	37 (6.0)	111 (28.9)	38 (15.3)	7 (6.7)
Transit improvement	10 (1.6)	13 (3.4)	47 (18.9)	0
Anti-diarrhea	2 (0.3)	6 (1.6)	2 (0.8)	16 (15.2)
With little/no lactose	13 (2.1)	14 (3.6)	11 (4.4)	12 (11.4)
Hypoallergenic formula	13 (2.1)	14 (3.6)	4 (1.6)	5 (4.8)
Formula not derived from cow’s milk	15 (2.4)	5 (1.3)	7 (2.8)	5 (4.8)
Treatment prescription—*n* (%)	357 (51.2)	306 (64.3)	201 (65.7)	77 (64.2)
Alginate, gastric protectant	157 (44.0)	72 (23.5)	21 (10.4)	23 (29.9)
Probiotic	123 (34.5)	166 (54.2)	72 (35.8)	49 (63.6)
Antisecretory drug	107 (30.0)	55 (18.0)	17 (8.5)	15 (19.5)
Antispasmodic drug	52 (14.6)	98 (32.0)	36 (17.9)	8 (10.4)
Laxative	26 (7.3%	19 (6.2)	102 (50.7)	0
Anti-diarrheal drug	5 (1.4)	5 (1.6)	1 (0.5)	16 (20.8)
Rehydration solution	7 (2.0)	7 (2.3)	4 (2.0)	11 (14.3)
Prokinetic drug	9 (2.5)	6 (2.0)	2 (1.0)	0

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
