# Peer review of "Functional Gastrointestinal Disorders in Outpatients Aged up to 12 Months: A French Non-Interventional Study"

_ijerph, 2020, doi:10.3390/ijerph17114031_

Round 1

Reviewer 1 Report

Whilst an attempt at characterising the nature and frequency of FGIDs in French infants is worthy and useful project, this study has some serious methodological and statistical flaws, which do not allow the conclusions within the manuscript to be made so strongly.

There is a significant selection bias in the infants included and also the data on symptoms is not more than parent recall.

There is much missing data, and consequently many of the conclusions cannot be drawn with any certainty.

Reviewer 2 Report

The authors investigate the effects of breastmilk vs. cow's milk on GI disorders in infants aged up to 12 months with equivalent numbers of subjects below and above 6 months.

The authors do a nice job of having a big population of infants from around France to evaluate indicating enough statistical power to make inferences from collected data. However, they miss a huge opportunity in not appropriately describing the social demographics of the population because as others studies have shown, this plays a huge role in infant diets and outcomes.

Additionally, it is not clear if the authors separate the effect of lower birth weight from incidence of FGID. It is not clear from this data that FGID is not merely an effect of low birth weight. It would help to know if there is no incidence of FGID in infants of low birth who were given breastmilk. 

Another issue is that there is no data on FGID issues being resolved after these infants were given probiotics; so it remains an assumption that it is changes in the gut microbiota that is causing these issues.

Lastly, these data show correlation and not causation. Unless the authors intent was to show correlations they cannot reach the conclusion they did. 

Minor issues:

Abstract: for acronyms, every word’s first letter has to be capitalized

There are no reference numbers, there is 0 in place of the number

Sentences repeated, lines 48 and 49.

Line 63, what is atopy?

Line 69, in US in infants

Table 1 needs to be reformatted

The numbers in Table 2 are wrong, the ones with FGID is 1,154 and not 1,1154

Reviewer 3 Report

Comments for the authors

The current study is a non-interventional, cross-sectional, and multicenter study which was conducted among private-ambulatory physicians in France. The aim was to estimate the incidence and the management of FIDGs in infants aged up to 12 months according the new ROME IV criteria and physician diagnosis. They concluded that the FGIDs are frequent in infants and associated with breast feeding and the earlier introduction of infant formula. In the context of the few publications on this subject (age less than 12 months), this study adds new information on the FGID epidemiology in France.

Major comments

  1. Abstract conclusions: “confirms the high frequency …. and the impact of breast feeding and the earlier introduction of infant formula”. The word “impact” should be replaced by “association”, because the study shows an association but does not confirm a causality that is implied by the “impact”.
  2. The references have not been cited in the text. Instead, the “zero” is written in the parentheses all over the text. Consequently, the relevance of the references with the respective text cannot be verified.
  3. The study design is well described and the statistical analysis is correct.
  4. The Tables 4-7 could be removed from the main text and submitted as supplemental file.
  5. Discussion

5.1.Although the beneficial effects of breast feeding on health cannot be disputed, the current study design does not allow clarification of whether the formula introduction preceded or followed the onset of FGID symptoms. It is not uncommon in cases with persisting FGID symptoms to change the breast-feeding to a special formula that could be beneficial for certain FGIDs. Therefore, one could consider the possibility that the higher prevalence of formula feeding after the first week of life and the earlier initiation of formula in the FGID group could reflect the physician’s effort to improve the symptoms of an emerging FGID (e.g. gastroesophageal reflux) rather than holding formula feeds responsible for the development of FGID symptoms. The authors could comment on this issue.

5.2.The statement in the last sentence “This large study also provided new information regarding the characteristics of infants having developed such digestive troubles suggesting the potential role of the gut microbiome”, is not supported by the design and results of the study. What the authors found is a lower incidence of breast feeding and earlier introduction of formula feeding in the FGID group. This finding does not necessarily suggest a role of the microbiome in FGID development considering the complicated interaction of breast feeding, gut microbiota, and gastrointestinal disorders. For example, it has been reported that the lower gut microbiota diversity and higher abundance of bifidobacteria, both of which are associated with breast-feeding, have opposite association with infantile colic prevalence.

5.3.A separate section of conclusions is missing

  1. The references in the text have been placed in parenthesis instead of square brackets and all are "0", while references in the list do not comply with the instructions for authors.
  2. The sections “author contribution” and “conflict of interest” are missing
  3. Left alignment is suggested for the 1st column of all Tables and for both columns of Table 1.

Minor comments

Line 29. “A recent literature review published in 2005 [0] reported …” the word “recent” should not be used for a publication dating back to 2005

Lines 48-49. The same sentence is repeated twice

Line 171. I suggest changing the “milk-fed” with “formula-fed”

Round 2

Reviewer 1 Report

The study readily confirms the burden of FGIDs presenting to paediatricians, and as has been concluded is certainly real world data.

The problem is that only children attending a paediatrician were included in this study, so population frequency is inherently biased when comparisons made between infants with and without FGIDs

Speculation on the role of the gut microbiome cannot be made on this data

Author Response

Point 1: The study readily confirms the burden of FGIDs presenting to paediatricians, and as has been concluded is certainly real world data. The problem is that only children attending a paediatrician were included in this study, so population frequency is inherently biased when comparisons made between infants with and without FGIDs. 

Response 1: We do agree with this comment, and this is the reason why this potential bias is already discussed in our manuscript: “However, most of the participating physicians were pediatricians leading to a potential selection bias as the infants they included may have been suffering from more severe disorders than those managed by GPs.” That being said, the frequencies we observed for infants’ FGIDs are consistent with previous findings from a recent multinational literature review and an expert consensus (Vendenplas 2015).

Point 2: Speculation on the role of the gut microbiome cannot be made on this data.

Response 2: We do agree with your comment regarding the potential role of the gut microbiota as it was not assessed in this cross-sectional study. However, as physicians frequently prescribed probiotics for infants with a FGID, we find it important to discuss the possible reasons for this prescription. However, the discussion on probiotics and gut microbiome has been shortened, the references adapted, and the potential role of the gut microbiome removed from the conclusion of our manuscript.